# Discrimination of Methicillin-resistant *Staphylococcus aureus* by MALDI-TOF Mass Spectrometry with Machine Learning Techniques in Patients with *Staphylococcus aureus* Bacteremia

**DOI:** 10.3390/pathogens11050586

**Published:** 2022-05-16

**Authors:** Po-Hsin Kong, Cheng-Hsiung Chiang, Ting-Chia Lin, Shu-Chen Kuo, Chien-Feng Li, Chao A. Hsiung, Yow-Ling Shiue, Hung-Yi Chiou, Li-Ching Wu, Hsiao-Hui Tsou

**Affiliations:** 1Institute of Biomedical Sciences, National Sun Yat-sen University, Kaohsiung 80424, Taiwan; pohsinkong@gmail.com (P.-H.K.); shirley@imst.nsysu.edu.tw (Y.-L.S.); 2Center for Precision Medicine, Chi Mei Medical Center, Tainan 71004, Taiwan; u104007417@cmu.edu.tw; 3Institute of Population Health Sciences, National Health Research Institutes, Zhunan, Miaoli 35053, Taiwan; chiang.louis@gmail.com (C.-H.C.); hsiung@nhri.edu.tw (C.A.H.); hychiou@nhri.edu.tw (H.-Y.C.); 4Institute of Precision Medicine, National Sun Yat-sen University, Kaohsiung 80424, Taiwan; 5National Institute of Infectious Diseases and Vaccinology, National Health Research Institutes, Zhunan, Miaoli 35053, Taiwan; ludwigvantw@gmail.com; 6Department of Medical Research, Chi Mei Medical Center, Tainan 71004, Taiwan; angelo.p@yahoo.com.tw; 7School of Public Health, College of Public Health, Taipei Medical University, Taipei 11031, Taiwan; 8Master’s Program in Applied Epidemiology, College of Public Health, Taipei Medical University, Taipei 11031, Taiwan; 9Graduate Institute of Biostatistics, College of Public Health, China Medical University, Taichung 40402, Taiwan

**Keywords:** methicillin-resistant *Staphylococcus aureus*, *Staphylococcus aureus* bacteremia, antimicrobial susceptibility testing, MALDI-TOF MS, machine learning, binning method

## Abstract

Early administration of proper antibiotics is considered to improve the clinical outcomes of *Staphylococcus aureus* bacteremia (SAB), but routine clinical antimicrobial susceptibility testing takes an additional 24 h after species identification. Recent studies elucidated matrix-assisted laser desorption/ionization time-of-flight mass spectra to discriminate methicillin-resistant strains (MRSA) or even incorporated with machine learning (ML) techniques. However, no universally applicable mass peaks were revealed, which means that the discrimination model might need to be established or calibrated by local strains’ data. Here, a clinically feasible workflow was provided. We collected mass spectra from SAB patients over an 8-month duration and preprocessed by binning with reference peaks. Machine learning models were trained and tested by samples independently of the first six months and the following two months, respectively. The ML models were optimized by genetic algorithm (GA). The accuracy, sensitivity, specificity, and AUC of the independent testing of the best model, i.e., SVM, under the optimal parameters were 87%, 75%, 95%, and 87%, respectively. In summary, almost all resistant results were truly resistant, implying that physicians might escalate antibiotics for MRSA 24 h earlier. This report presents an attainable method for clinical laboratories to build an MRSA model and boost the performance using their local data.

## 1. Introduction

Early administration of broad-spectrum antimicrobial agents is one of the most important steps in the management of septic patients [1,2]. Kumar et al. (2006) reported a 7.6% decrease in survival per hour of delay of antibiotic therapy [3]. Seymour et al. (2017) reported that the odds of in-hospital death were 14% lower in patients receiving antibiotics within 3 h than those treated 3 to 12 h after the recognition of sepsis [4]. Gram-positive cocci are taking an increasingly significant role as major etiologic agents of bacterial sepsis [5]. *Staphylococcus aureus* accounts for 20% to 30% of culture-positive infections and 20% of bloodstream infections among intensive care unit (ICU) patients [6,7]. Furthermore, *S. aureus* bacteremia (SAB) has been associated with mortality rates of 34.4% and 18.9% in ICU and non-ICU patients, respectively [8]. In addition to the high incidence of SAB, antibiotic resistance is emerging as a major clinical challenge. The prevalence of methicillin-resistant *S. aureus* (MRSA) varies widely between geographic regions, from <1% to >50% [9,10,11,12,13].

Empiric therapy of suspected MRSA infections requires consideration of the resistance profile of local strains. Vancomycin or daptomycin are usually recommended as first-line antibiotics [14], while linezolid, teicoplanin, or other agents are alternatives for treating susceptible isolates [15]. The outcome of MRSA bacteremia is worse than that caused by methicillin-susceptible *S. aureus* (MSSA); Cosgrove et al. (2003) found a pooled odds ratio of mortality of 1.93 (95% CI, 1.54–2.42) [16]. Because of the high prevalence of MRSA infections, clinicians consider empiric broad-spectrum antibiotic therapies that usually include vancomycin. However, inadequate empiric or unnecessarily broad-spectrum regimens are associated with higher mortality and risks of colonization and superinfections due to antibiotic-resistant microorganisms [17,18]. Furthermore, empiric anti-MRSA treatment of MSSA bacteremia has been associated with worse outcomes in comparison with standard therapies [19,20,21]. Therefore, early recognition of MRSA infection is critical for the appropriate use of anti-MRSA therapies.

Routine antimicrobial susceptibility testing (AST), based on bacterial growth in the presence of specific antimicrobial compounds, requires 18 to 24 h after species identification [22]. Recently, more rapid assays have attained regulatory approval, but most require additional equipment and manipulations [23,24,25]. In contrast, matrix-assisted laser desorption ionization with time-of-flight (MALDI-TOF) mass spectrometry (MS), already implemented widely for routine species identification [26,27], yields comprehensive proteomic information that may disclose potential biomarkers of antibiotic resistance [28,29]. Discrimination of resistant strains by characteristic MALDI-TOF MS mass peaks is not novel. Some studies have reported sensitivity and specificity in the discrimination of MRSA by using single or several mass peaks, especially for the peak proposed between the mass-to-charge ratio (m/z) 2411 to 2415, originating from a small peptide termed PSM-mec that is highly specific to *mecA*-positive MRSA strains [30,31,32,33]. However, a single mass peak or peak combination usually corresponds to a subset of resistant strains. For example, 15.7 to 23.2% of MRSA display a peak between m/z 2411 to 2415 [28]. To improve the discriminatory performance and to extend the coverage of resistant strains, machine learning (ML) techniques have been incorporated to investigate mass data comprehensively. Tang et al. (2019) examined 10 MRSA and 10 MSSA isolates with both supervised and unsupervised ML, demonstrating over 90% accuracy [34]. Wang et al. (2021) tested ML algorithms on a large collection of 2500 MRSA and 2358 MSSA clinical isolates, reporting 77% sensitivity and 77% specificity [35].

In addition to MRSA, other multi-drug-resistant pathogens have also been explored. Peaks at m/z 11109 are correlated to a beta-lactamase variant and discriminate some strains of carbapenem-resistant *Klebsiella pneumoniae* (CRKP) [36,37]. For example, Huang et al. (2020) classified 46 CRKP and 49 CSKP isolates with an accuracy of 97% [38]. However, a limitation of the ML technique is the unclear association between particular resistance mechanisms and discriminating peaks, which might only identify a particular clone rather than drug resistance [28]. Therefore, enrollment of an external dataset is suggested to evaluate the effect of clonality [39]. The establishment of regional models with thorough validation would be appropriate [35].

In this study, a modified binning method was developed to preprocess MS data, and four ML models, i.e., support vector machine (SVM), decision tree (DT), random forest (RF), and polynomial regression (PR), were constructed to classify MRSA and MSSA. Genetic algorithm (GA) was applied to search optimal parameters for modeling, and the performance improved markedly in comparison with the default parameters. Models were trained by six-month clinical samples isolated from bloodstream culture and validated independently by the samples isolated during the following two months. The independent testing indicated that SVM were the best model for the classification, coming out with accuracy, sensitivity, and specificity of 87%, 75%, and 95%, respectively. These results showed a possible extension of MALDI-TOF MS for the early diagnosis of MRSA infections in clinical practice.

## 2. Materials and Methods

### 2.1. Collection of Clinical Data and Sample Annotation

The study design was reviewed by the Institutional Review Board of the Chi Mei Medical Center and followed the approved process to collect clinical and MS data. All data were derived from our hospital-constructed database, in which data had been unlinked and anonymized. Inclusion criteria were: (1) blood samples obtained during December 2020 through July 2021; (2) positive blood culture; (3) species identified as *S. aureus*. The annotation of MRSA or MSSA was based on oxacillin resistance in AST, represented by “−1” and “+1”, respectively. There were 366 samples, obtained from December 2020 to May 2021, applied for feature extraction and construction of ML models, and 182 samples from June and July 2021 were set as an independent testing dataset for evaluating model performance. The workflow of this study is shown in Figure 1, and the tabular data in Figure 1 are presented in Appendix A.

The methods for species identification and AST followed the standard protocols of our clinical microbiology laboratory as previously published [40]. Briefly, positive blood-cultured samples were subjected to a plate culture overnight, and bacterial colonies were picked and spotted on a target plate of MALDI-TOF MS (MALDI Biotyper, Bruker Daltonik, Bremen, Germany). After being treated with formic acid and alpha-cyano-4-hydroxycinnamic acid, the mass spectra were acquired and interpreted by comparing with the database (MALDI Biotyper Library, Bruker Daltonik, Bremen, Germany), and the software generated the score of the best match. Matching scores less than 1.7, 1.7 to 2.0, and greater than 2.0 were considered as no organism identified, low-confidence, and high-confidence, respectively. Only results with matching scores greater than 1.7 were acceptable for clinical usage and thereby enrolled to this study.

Oxacillin resistance was determined by disk diffusion method and interpreted the result according to Clinical and Laboratory Standards Institute (CLSI) [41]. The bacterial colonies were resuspended in Tryptic Soy Broth (BD Biosciences, Franklin Lakes, NJ, USA) to 0.5 McFarland turbidity and inoculated to a Mueller–Hinton agar plate (Thermo Fisher Scientific, Waltham, MA, USA), and appropriate antibiotic disks were placed afterward. After 24 h of incubation, resistant characteristics were revealed by measuring the zone diameter of growth inhibition. For MRSA, the test of oxacillin resistance is surrogated by the zone diameter of a 30 ug cefoxitin disk within 21 mm [41].

### 2.2. Feature Extraction

Raw data from MS must be preprocessed to a structured data matrix with resistance labels. Mass spectra were converted to text format containing peak information such as m/z, full width at half maximum (FWHM), signal-to-noise ratio (S/N), peak area, intensity (IN), and relative intensity (RI) by flexAnalysis 3.4 (Bruker Daltonik, Bremen, Germany). 

Because m/z and total number of peaks change often between MS scans, these data are usually not directly applicable to ML. To address this variation, binning is commonly used to organize MS data by dividing continuous m/z values into several bins [34,35]. Each bin is considered as a range of m/z; in other words, any peak located within the range would be attributed to the same feature for modeling. For example, peaks located between m/z 6583.412 and 6601.764 would be attributed to the feature designated 6593.2. However, binning may cause problems when applied to new samples because their set of peaks may not match existing bins; therefore, a list of reference peaks with specific bins must be generated first, so that the peaks of new samples may be attributed to features accordingly.

We used the binning method but modified the strategy of setting bin size by automatically adjusting bin size by peak width. To construct the list of reference peaks with specific bins first, all raw peaks from the training samples allocated for model construction were combined, and the gap between the smallest and next smallest m/z was calculated. These two peaks would be set as the same feature if the gap was not greater than 0.5-fold FWHM; otherwise, a new feature would be provided. After iterating all peaks from smallest to greatest m/z, a list of reference peaks was constructed; the lower and upper bounds of each bin corresponded to the peaks at the lowest and the highest m/z in a feature. The name of a reference peak was represented by the average among m/z of peaks in a feature. With the list of reference peaks, the peaks of the training sample set and the testing sample set were separately extracted to two data matrices. The data value of a peak located within a bin was extracted to the feature, whereas the data value was assigned as “0” if no peak was presented in the bin. 

There were four types of data value employed to represent peak abundance, namely S/N, IN, RI, and total ion current-normalized intensity (the individual signals (TIC) in a spectrum were divided by the sum of signals). These four data types were compared and evaluated by ML modeling.

Feature extraction was evaluated by using different sample sizes. We randomly selected 36, 72, 144, 216, and 288 samples from the training sample set to build different sets of reference peaks and observed the numbers, variations, and intersections in these sets.

### 2.3. Construction of Machine Learning Models

Four ML models, i.e., DT [42], PR [43], RF [44], and SVM [45], were applied to classify MRSA and MSSA and were built by the scikit-learn package [46] of Python programming language. The main tasks performed on these four ML models were: (1) optimizing the ML models; and (2) using the independent samples to verify the classification performance. Task (1) consists of two steps: (i) the use of Gini impurity [47] in the DT algorithm to find important features; and (ii) the use of these important features to revise the data matrix and apply a genetic algorithm (GA) to search for the optimal parameters of the ML models. GA is a computational model of biological evolution, imitating Darwin’s rule “Natural selection, survival of the fittest”, and it is useful both as a search method for solving optimization problems and for modeling evolutionary systems [48]. In each iteration of evolution, the GA generates a new population and replaces the preceding one through the genetic operations of selection, crossover, and mutation. After continuous evolution of the GA, the quality of the solutions improves. Another advantage of GA is its reduction of search time. To find optimal parameters, some studies [35,49,50] used grid search (an exhaustive search), which is comparatively time intensive. The DT, RF, and SVM are classification algorithms that generate a class as a result, whereas PR, a regression algorithm, returns a numerical value as a result. An isolate would be classified as MRSA for PR < 0, or as MSSA for PR ≥ 0.

### 2.4. Important Feature Selection

We use the attribute “*feature_importances*” in the DT to calculate the importance for each feature. Thus, before calculating feature importance, we must build the DT for the training samples. For a decision tree, scikit-learn calculates a node importance, denoted as *NI_j_* for node *j*, using Gini impurity [47]. Suppose a binary tree has only two child nodes, and the importance of node *j* is [51]
*NI_j_* = *w_j_C_j_* − *w_Lj_C_Lj_* − *w_Rj_C_Rj_*,(1)
where *w_j_* is the weighted number of samples reaching node *j*. *C_j_* is the impurity value of node *j*. *L_j_* and *R_j_* indicate the child node from the left and right split on node *j*, respectively. The Gini impurity of node *j*, *C_j_*, is calculated by [52]
(2)$$C_{j}=1-\sum_{i=1}^{2}p_{i}^{2},$$
where *p_i_* is the fraction of items labeled with class *i* in the set. For example, suppose there 13 samples in class ‘1′, and 20 samples in class ‘0′. Thus, *p*_1_ = 13/33 and *p*_0_ = 20/33. There are only two classes of a binary tree here.

The importance for each feature is calculated by [51].
(3)$$FI_{i}=\frac{\sum_{j:node\;j\;splits\;on\;feature\;i}NI_{j}}{\sum_{k\in all\;nodes}NI_{k}}$$
where *FI_i_* is the feature importance of feature *i*, and *NI_j_* and *NI_k_* are calculated by Equation (1). *FI_i_* can be normalized into [0, 1], as calculated by
(4)$$normFI_{i}=\frac{FI_{i}}{\sum_{j\in all\;features}FI_{j}}$$
where *normFI_i_* is the normalized Feature Importance of feature *i*.

### 2.5. Performance Measurement

ML model performance was assessed by accuracy, sensitivity, specificity, and Area Under a receiver operating characteristic (ROC) Curve (AUC). The horizontal axis of the ROC curve is the false positive rate (FPR), equal to 1—specificity, and the vertical axis is the true positive rate (TPR). A trapezoid is formed between two false positive rate points, and thus, AUC is calculated by the sum of the area of all trapezoids. Definitions are shown below:(5)$$\mathrm{Accuracy}=\frac{\mathrm{TP}+\mathrm{TN}}{\mathrm{TP}+\mathrm{TN}+\mathrm{FP}+\mathrm{FN}},$$
(6)$$\mathrm{Sensitivity}=\frac{\mathrm{TP}}{\mathrm{TP}+\mathrm{FN}},$$
(7)$$\mathrm{and}\;\mathrm{Specificity}=\frac{\mathrm{TN}}{\mathrm{TN}+\mathrm{FP}}.$$
where TP, TN, FP, and FN stand for true positive, true negative, false positive, and false negative, respectively.
(8)$$\mathrm{AUC}=\sum_{i=1}^{n-1}(tpr^{i}+tpr^{i+1})\cdot \left(fpr^{i+1}-fpr^{i}\right)/2,$$
where *n* is the number of elements in the FPR data set, *tpr^i^* denotes the *i*-th data point of TPR, and *fpr^i^* is the *i*-th data point of FPR.

### 2.6. SCCmec-Typed MRSA Isolates

To further clarify the bias of the ML models, classification accuracy was assessed by another 110 Staphylococcal Cassette Chromosome *mec* (SCC*mec*)-typed MRSA isolates. All these strains, generously provided by Infectious Diseases Laboratory, Department of Medical Research, Chi Mei Medical Center, had been typed by multiplex PCR in other study [53] and stored at temperatures below −70 °C. Preserved bacterial stocks underwent plate subcultures to recover viability before MALDI-TOF MS analysis. The rest of the procedure is the same as that for clinical samples.

## 3. Results

### 3.1. Evaluation of Feature Extraction

A total of 26,159 peaks were obtained from the training sample set, and 508 reference peaks were extracted by the modified binning method. A data matrix with 366 (147 MRSA and 219 MSSA isolates) × 508 (features) was used for model construction, and another data matrix with 182 (72 MRSA and 110 MSSA) × 508 (features) was applied for independent testing. Because the 508 reference peaks were generated from the training samples, some peaks in the testing samples were inevitably lost because they were not fit for the bins of reference peaks. For example, a peak at m/z 2166.273 was observed in an independent sample, but the two closest reference peaks ranged from 2149.044 to 2157.791 and from 2166.364 to 2172.865. To address the influence of peak loss, we first evaluated whether 366 samples were enough to construct reference peaks by comparing feature numbers generated from different sample sizes. More than 50% of reference features were generated by one-tenth of the total samples (Table 1), and the trend of feature numbers increased with sample size but slowed while approaching 366 samples, indicating that 366 samples might be adequate for the derivation of reference peaks. In addition, the variations of m/z in the top 10 important features generated by DT were reviewed; for example, the mean m/z of the most important feature, m/z 6593.2, varied only from 6593.2 to 6594.2, showing that the robustness of the feature extraction process was acceptable.

### 3.2. The Performance of Important Features

This study investigated the classification performance of the four ML models for four different data sets as described below under three feature selection strategies.

There was a total of 366 samples in each data set, and each sample had 508 features. This study used one of the functions, i.e., *feature_importances_*, in the decision tree algorithm of the scikit-learn package of Python to calculate the Feature Importance of each feature. We repeated the experiment 100 times, and conducted the following procedures for each experiment:From 366 samples, 80% of the samples were randomly selected as training samples, and the remaining 20% were used as testing samples.A decision tree model was constructed based on the training samples as mentioned in (1), and Feature Importance was calculated for each feature.There were three strategies to select features for each sample in the training and testing samples mentioned in (1).
(3-1) Among the 508 features, only those whose Feature Importance were greater than 0 were retained, and remaining features were deleted.(3-2) Only the top 15 features in terms of Feature Importance were kept, and the remaining features were deleted.(3-3) All 508 features were retained.The four ML models were built based on the training samples with the features selected in the above (3-1) to (3-3). The classification performances of the four ML models for items (3-1) to (3-3) were evaluated.

The experimental results for the S/N data set are shown in Table 2. For using the scikit-learn package of Python, the parameter setting for the SVM model is kernel function: RBF (Radial Basis Function), *C* = 100, and *γ* = 3.4425 × 10^−5^. The highest polynomial order of PR model is 1. DT and RF models use the default parameters. The words “important feature” as shown in Condition 1 and 2 of Table 2 indicate a feature whose Feature Importance is >0.

According to the training samples as mentioned in item (1), there were about 31 features with an average Feature Importance > 0. As shown in Table 2, the performance of training samples was better than the performance of testing samples in all four ML models. Moreover, the accuracy, sensitivity, and specificity of the training samples of DT and RF models were almost 100%. Table 2 shows that specificity performance was almost always better than sensitivity performance. Classification accuracies using all important features were higher than those obtained by using only the top 15 important features. SVM had the highest classification accuracy of testing samples that used all features; however, DT, RF, and PR methods had the highest accuracy of testing samples that used all important features.

The classification performances of the testing samples of the four ML models for different data sets under different feature selection strategies are shown in Figure 2. The experimental procedures in Figure 2 are the same as those in Table 2. The accuracies of the four ML models using different data sets under the three feature selection strategies were consistent. For all data sets, the SVM model had the best accuracy when using all 508 features, as shown in Figure 2A. However, the RF model had higher accuracy with the strategy of selecting the top 15 important features (Figure 2G). For the PR model, the strategies of selecting the top 15 important features and selecting all important features had higher accuracy (Figure 2J). For the DT model, the classification accuracies of the three feature selection strategies were similar (Figure 2D).

The SVM model had the highest sensitivity and specificity when using all 508 features (Figure 2B,C). In the following experiments of parameter optimization, the 80% of the S/N data set with 366 samples was used to build the training and testing samples.

### 3.3. Parameter Optimization of Machine Learning Models

To save time, this study used GA to find acceptable and satisfactory rather than optimal parameters of SVM, DT, and RF models.

#### 3.3.1. Parameter Selection of Polynomial Regression Model

In the PR model, only one parameter needed adjustment, i.e., the *degree* of the PR model. As shown in Table 2, the performance of using important features was better than that of using all features. As shown in Appendix A, for the performance of the testing samples with a degree of 1 was best. In the performance of testing samples, the highest classification accuracy and specificity were 0.7638 and 0.8596 (using the top 15 important features); the highest sensitivity was 0.6245 (using all important features). When all important features were used to classify and the degree of PR model was greater than 1, the classification accuracy, sensitivity, and specificity of the training samples were all 1.0. However, the classification accuracy, sensitivity, and specificity of the testing samples were relatively low (almost all lower than 0.6). This demonstrates the phenomenon of overfitting. According to the experimental results, the optimal parameter of the PR model is degree = 1.

#### 3.3.2. Parameter Selection of Support Vector Machine Model

GA was used to search the optimal parameters of SVM, including *C* and *γ* parameters, and the kernel function was set as ‘RBF’. The chromosome design and the definition of fitness function of a chromosome of GA are shown in the text of the Supplement and in Figure 3. To optimize the parameters of SVM, we set GA to have 60 chromosomes, with each chromosome encoding 25 genes. The parameter settings of GA are shown in Appendix A. The maximum evolutionary iteration of GA is 200, and the fitness values during the evolution are shown in Appendix A. The maximal fitness value during the evolution was 0.8021, and the best 10-fold cross-validation accuracy was 0.8187. The optimal parameters were: kernel = ‘RBF’, *C* = 719.66, *γ* = 0.0001, and the number of features was 13 (the top 13 features among all the important features found by DT model).

#### 3.3.3. Parameter Selection of Decision Tree Model

GA was used again to search for the optimal DT parameters. This study used the DT model of the scikit-learn package of Python (scikit-learn, 2021). We applied GA to investigate the following parameters: criterion (Cr), splitter (Sp), max_depth (MD), min_samples_split (MSS), min_samples_leaf (MSL), ccp_alpha (CA), and *N_fea_* (the number of features in the samples).

The chromosome design of GA for searching best parameters of DT is shown in Figure 4. This study used 35 binary bits to encode seven parameters. The parameter settings of GA are listed in Appendix A. After 200 generations, GA found that parameters of DT with acceptable 10-fold cross-validation accuracy, i.e., 0.8011, and the corresponding fitness values was 0.7933. The found parameters are Cr = ‘gini’, Sp = ‘best’, MD = 15, MSS = 19, MSL = 10, CA = 0.0172, and *N_fea_* = 7 (the top 7 features among all the important features found by DT model). The maximal and average fitness values during evolution are shown in Appendix A.

#### 3.3.4. Parameter Selection of Random Forest Model

Because RF is composed of multiple DTs, many parameters of RF and DT are identical. As with the DT parameters described above, this study deleted the first two parameters (Cr and Sp) and added two RF parameters, i.e., bootstrap (BS) and n_estimators. The BS parameter is employed to decide whether BS samples are used to build samples. If it is false, the whole dataset is used to build each tree. n_estimators (NE) indicates the number of trees in the forest. In this study, GA is still used to search for the optimal parameters of RF. Here, the chromosome design of GA is like Figure 4. However, the first gene representing Cr in Figure 4 is changed to represent BS parameter; the second gene that represents Sp is replace by NE parameter encoded by six binary bits. The parameter settings of GA are listed in Appendix A. The optimal fitness value after 200 generations was 0.8040 and its 10-fold cross-validation accuracy was 0.8121. The found parameters were BS = True, NE = 170, MD = 81, MSS = 5, MSL = 1, CA = 0.000098, and *N_fea_* = 5 (the top five features among all the important features found by DT model).

### 3.4. Performance of Independent Testing

Table 3 show the classification performance of the four ML methods with or without optimal parameter settings. This study used 366 samples of patient data from December 2020 to May 2021 to train the ML methods and used 182 samples from June 2021 to July 2021 for independent testing. Table 3 is divided into two parts. In the upper part, the ML methods used the optimal parameters in the models, and the training and testing samples had small numbers of important features (smaller than 30). The PR model was trained and tested using the top 15 important features. In the second half, the ML methods use the default parameters of scikit-learn package in the model, and all 508 features are used in the training and testing samples.

The investigation demonstrated that the ML models trained by the optimal parameters and using the selected features could provide better performance, especially for the SVM model in which the AUC value increased from 0.5910 to 0.8664 (accuracy increased from 0.5879 to 0.8736). In the independent testing, the SVM model performed best, with accuracy, sensitivity, specificity, and AUC of 0.8736, 0.7500, 0.9545, and 0.8664, respectively. The assessment of accuracy, sensitivity, and specificity and the evaluation of AUC were performed at different times. The ROC curves of the independent testing for the four ML models with and without optimal parameter settings are shown in Figure 5.

### 3.5. Verification by SCCmec-Typed MRSA

A total of 110 MRSA isolates carrying four types of SCCmec were investigated to clarify bias against different stains. By using the best models trained by clinical samples, the accuracy varied in different SCCmec types. The models were able to recognize most type III and more than half of type IV and V isolates, but nearly none for type II (Part 1 of Table 4). A possible explanation for this bias was clonality of the training samples, which means the number of type II strain in training samples might be too small to train models, and this assumption was also supported by some studies surveying the prevalence of SCCmec in Taiwan revealed that type III was most prevalent, followed by type IV or V and type II at last [54,55,56]. To address this bias, we manually set the importance of feature 2410 to 2417, which was widely reported as characteristic feature of type II [30,31,32,33], to top priority, and the accuracy improved dramatically (Part 2 of Table 4). Except RF, the other three models recognized over 90% MRSA in the four types of SCCmec. Unfortunately, this manual modification did not improve but slightly decline the classification performance in the independent testing samples (Appendix A), implying differences between clinical samples and standardized isolates.

## 4. Discussion

Feature extraction is a critical step in data preprocessing, and binning strategy is an efficient method for organizing MS data. This strategy is required to determine bin size before processing; for instance, Wang et al. (2021) explored bin size ranging from 1 to 15 m/z, and found that the best classification performance was obtained when using 10 m/z [35]. However, a fixed bin size might carry disadvantages: (1) a peak located in a junction of bins would be taken apart into two features; (2) the problem of mass shifting would be more obvious when scanning high mass range in MALDI-TOF MS; consequently, a wider bin may be needed to cover a feature at high m/z. In this study, we developed a modified binning method, in which bin size was adjusted according to the peak width of training samples, and that generated a list of reference peaks. Peaks in new samples could be converted to features of the current model, making it feasible for subsequent applications.

The types of data value to represent feature intensities may influence classification performance, and S/N seems to be the best measure in comparison with IN, RI, TIC in this study (Figure 2). Generally, mass spectrum intensities are presented by either absolute abundance, such as peak area and peak height, or relative abundance, usually normalized by base peak or TIC. However, the reproducibility of peak abundance is usually poor due to the crystallization nature of MALDI [57]. In addition to the abundance of peak, S/N has been demonstrated as an alternative measure to represent signal intensities and is useful for quantitation of certain biomolecules in biological samples with acceptable linearity [58,59]. Studies using MALDI-TOF MS to discriminate antibiotic-resistant strains disagree on the types of data values. Huang et al. (2020) used intensity versus m/z to present their spectral data [38], Wang et al. (2021) used z-score normalization for each spectrum [35], while Tang et al. (2019) found that peak area was better than peak height for the classification performance assessment [34]. Though the demands on the relationship between amount and signal for ML are not as strict as for quantitative assays, feature intensities, and especially important features, may affect classification results. Therefore, variability of intensity may degrade model performance; consequently, we recommended exploration of the influence of data type during model construction.

Table 2 and Figure 2 disclose that the SVM had the best performance for classification using all 508 features. However, the classification accuracy of PR and RF methods that use all features to classify was worse than that of using only important features. We discuss this problem from the principles of these models. Figure 2A shows that the accuracy of SVM was highest when all features were used for classification, but the parameter optimization result shows that the optimal number of features is *N_fea_* = 13. The possible reason for the inconsistency between the two is that the parameters of SVM, i.e., *C* and *γ*, are set differently, which will lead to different optimal number of features. The SVM parameters of Figure 2A use *C* = 100 and *γ* = 3.4425 × 10^−5^, and the optimal parameters found after GA evolution were *C* = 719.66 and *γ* = 0.0001.

First, SVM seeks to find a linear or non-linear boundary (the linear boundary is a hyperplane) to distinguish different classes of data [60]. Only by using all the features in the training sample can this boundary be accurately defined. If the number of features in the sample is reduced, the features considered by the boundary will also be reduced (the dimensionality of the boundary will be reduced). Consequently, the classification accuracy of the boundary will decrease.

Secondly, PR is a special case of multiple linear regression [43]. In the backward stepwise regression procedure: First, consider all independent variables to establish a complete linear regression model. Secondly, removing a variable causes the Akaike Information Criterion (AIC) to drop the most [61]. Repeat this step until the AIC value can no longer drop. Therefore, considering fewer independent variables may improve the prediction effect of the model. The classification performance of PR is better when considering only important features rather than considering all features, which may be like the above-mentioned principle.

Third, RF is composed of several decision trees. When the number of considered features increases, the decision tree is larger, and the number of rules is greater. Subsequently, the classification efficiency of the training samples is higher, but the efficiency of the testing samples is relatively poor. This is the phenomenon of overfitting, that represents an over-adjustment of parameters to suit the training samples and may therefore fail to fit additional data or predict future observations reliably [62]. Therefore, considering only important features can reduce RF overfitting and improve its classification performance.

Table 2 demonstrates that the PR model has obvious overfitting. Figure 2J,L clearly indicate that the performance of PR using all features for classification was worse than that of using only important features. To overcome overfitting, we can consider using only the important features of the samples.

The use of GA requires a prior determination of the number of chromosomes (population size) and the number of genes in each chromosome. The number of genes depends on the encoding method of the parameters, which are solutions of a problem, and can be determined according to the needs of the problem. For an example, please refer to Supplemental Figure 1. The greater the population size, the easier it is to find the optimal solution; however, the process takes more computing time. The determination of the population size and the number of genes can rely on professional experience or experimental results or may refer to the design methods mentioned in relevant literature [63,64,65].

The classification performances of the training samples of the SVM and PR models were like the performances of the testing samples (Table 3). However, the classification performances of the training samples of the DT and RF models were better than the performances of the testing samples. Therefore, the classification performances of SVM and PR models were relatively robust under optimal parameters.

Chung et al. developed a web tool for the rapid identification of oxacillin-, clindamycin-, and erythromycin-resistant *S. aureus* [66]. They also used MALDI-TOF MS with ML, i.e., DT, RF and SVM, models to classify whether these *S. aureus* bacteria were resistant. Feature importance scores calculated by RF, whereas we calculated by DT, to find the appropriate number of features to classify the three antibiotic resistances separately. The features were ranked by their important scores, and then added in the model sequentially until the accuracy reached a plateau. They selected 36, 38, and 37 features for oxacillin, clindamycin, and erythromycin models, respectively. Instead, we applied the GA to find the optimal number of features under a certain combination of parameters. As discussed earlier, SVM has different optimal number of features for different parameter combinations. Thus, the number of features found Chung et al.’s study may not fit optimal parameter combination. In addition, Chung et al. used the grid search to find the optimal parameters for ML models [66]. When the model has many parameters, the grid search may require a lot of computation time. In the study by Chung et al. (2021), the numbers of data in training set and independent test set for constructing multiple antibiotics resistance model were 15,689 and 3554. In contrast, the number of samples we used for training ML models and independent testing were only 366 and 182, respectively. In the study by Chung et al. (2021), the performance on the independent test set under the optimal parameters were 0.7918 (sensitivity), 0.9053 (specificity), 0.8545 (accuracy), and 0.8486 (AUC). In our study, the sensitivity, specificity, accuracy, and AUC of the independent testing of SVM under the optimal parameters were 0.7500, 0.9545, 0.8736, and 0.8664. The above results show that this study can obtain satisfactory classification results even when using fewer samples to train the ML model.

Our results demonstrated that parameter optimization has a very significant impact on model performance. These ML methods offer their best performance after parameter optimization in different applications. Because of different applications, the data patterns are different. The parameters of the ML models must be adjusted to suit the data patterns for a specific application. The classification performances of SVM, RF, and PR models improved significantly when optimal parameter settings were used (Table 3). Besides, the model performance verified by SCC*mec*-typed MRSA pointed out it is important to establish ML models by regional samples to avoid differences between training and testing samples, and post-implementation evaluation and recalibration would be needed because the prevalence of genotypes might change from time to time.

Although the model results were not identical to those of AST, we propose the model is still worth to incorporate into clinical workflow. Once a model is established, the only additional task for a laboratory staff is taking few minutes to operate a computer to analyze sample data. In comparison, most rapid assays, no matter genotypic or phenotypic methods [23,67], may request couples of hours for manipulations of samples or instruments. Moreover, the result of our best model is 75% sensitivity and 95% specificity in independent testing; in other words, not all resistant isolates were recognized by the model, but almost all isolates classified as resistant were truly resistant. Therefore, a positive result would represent an indication for anti-MRSA therapies, shortening 18 to 24 h, whereas a negative result does not rule out MRSA infection. Their current therapies would be kept and pending for AST results. Consequently, the incorporation into current laboratory workflows is highly feasible with minimal cost and labor time, and the treatment of MRSA may start almost 1 day earlier than before.

## 5. Conclusions

To our knowledge, this study is the first example of the use of a modified binning method to extract features. Four ML models were tested to classify MRSA and MSSA, and parameters were searched by GA. The classification performance was markedly improved after parameter optimization. The classification results of independent testing showed that SVM was the best model, with accuracy, sensitivity, and specificity of 87%, 75%, and 95%, respectively, and an AUC of ROC curve of 0.8662. In addition to acceptable model performance, independent testing proved that the process and ML model were applicable to new samples as in real-life situations in clinical laboratories. Although thorough validation is required before clinical implementation, these results may facilitate studies of whether clinical outcomes may be enhanced by the early discrimination of MRSA.

## Figures and Tables

**Figure 1 pathogens-11-00586-f001:**
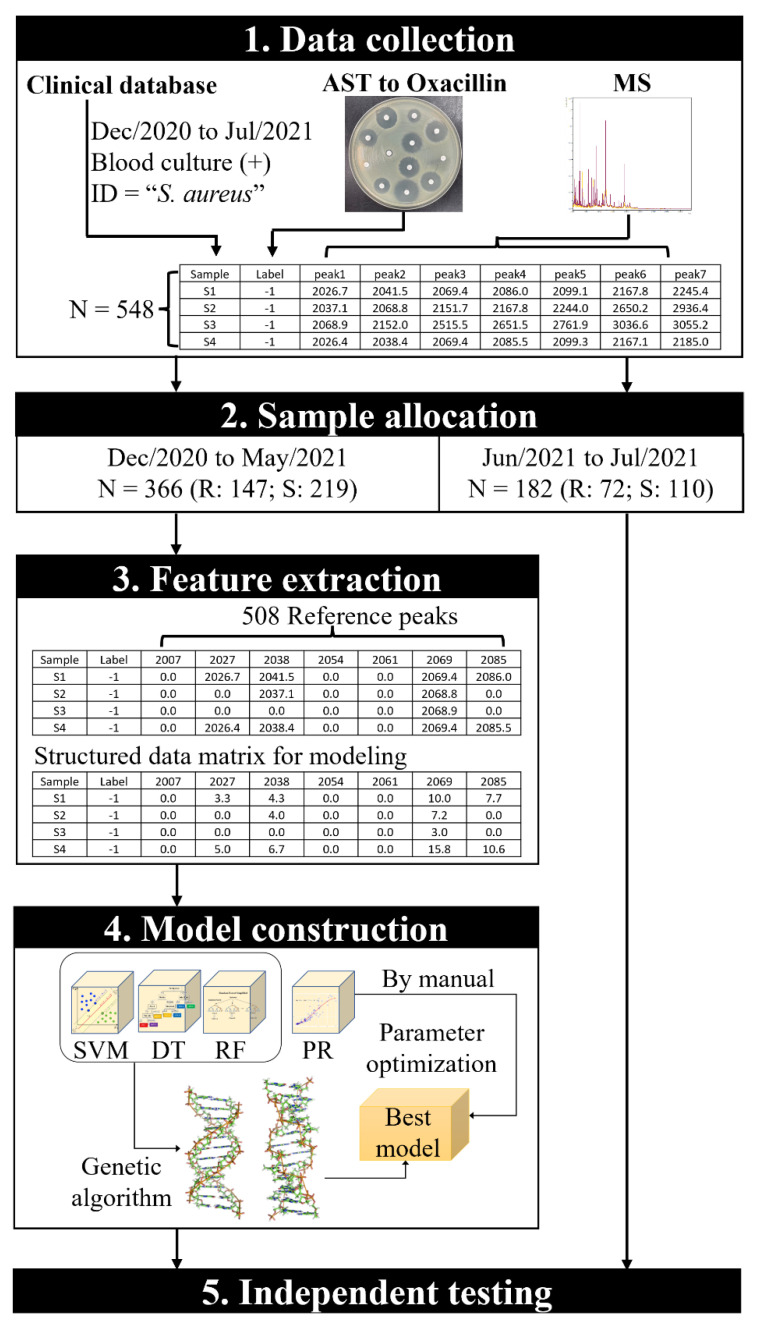
The workflow of this study.

**Figure 2 pathogens-11-00586-f002:**
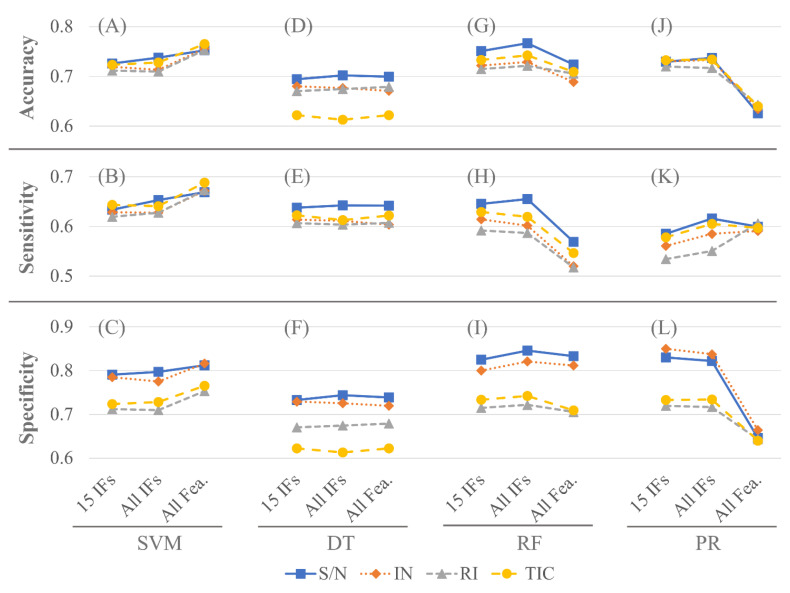
The classification performance of different methods, i.e., SVM (**A**–**C**), DT (**D**–**F**), RF (**G**–**I**), and PR (**J**–**L**), in different feature selection strategies. The horizontal axis represents the number of features. 15 IFs: top 15 important features, All IFs: all important features, All Fea.: all 508 features.

**Figure 3 pathogens-11-00586-f003:**
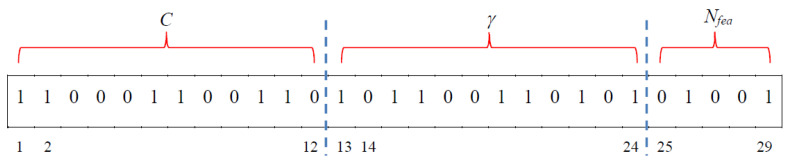
An illustration of chromosome design for optimize SVM parameters. The chromosome is encoded as binary bits. Genes 1 to 12 (*N_bit_* = 12) are used to represent the parameter *C*; genes 13 to 24 are used to represent the parameter *γ*; genes 25 to 29 are used to represent the number of features *N_fea_* of the sample.

**Figure 4 pathogens-11-00586-f004:**
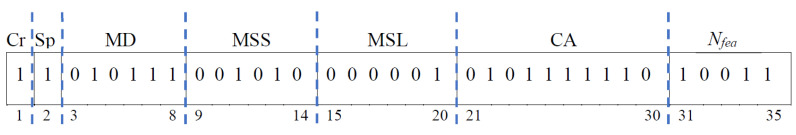
An illustration of chromosome design for optimize DT parameters. The chromosome is encoded by 35 binary bits. The full terms for the abbreviated parameters are: Cr—criterion; Sp—splitter; MD—max_depth; MSS—min_samples_split; MSL—min_samples_leaf; CA—ccp_alpha; *N_fea_*—number of features.

**Figure 5 pathogens-11-00586-f005:**
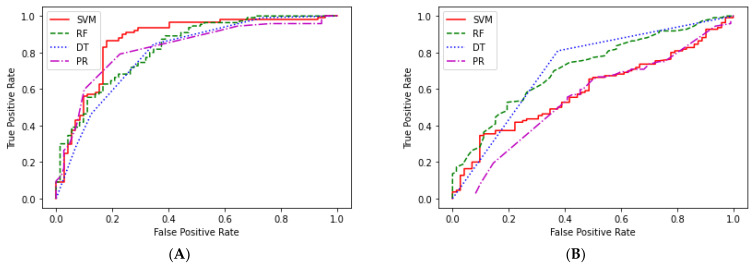
The ROC curve of the independent testing. (**A**,**B**) are the ROC curves for the four ML methods with and without the optimal parameter settings.

**Table 1 pathogens-11-00586-t001:** Evaluation of feature extraction by repeated random sampling from 366 samples.

Sample No.	36	72	144	216	288	366		
Total feature No.	290	324	424	429	461	508		
**Importance ***	**Mean m/z Calculated by Different Sample Size**						**Mean**	**SD #**
0.184	6594.2	6593.4	6593.5	6593.3	6593.2	6593.2	6593.5	0.36
0.093	5305.0	5304.8	5305.0	5304.8	5304.8	5304.8	5304.9	0.12
0.078	7420.9	7420.7	7420.8	7420.4	7420.7	7420.8	7420.7	0.19
0.071	6425.1	6424.7	6425.3	6424.8	6424.8	6424.8	6424.9	0.23
0.061	13183.3	N/A	13185.3	13186.5	13184.7	13185.3	13185.0	1.17
0.049	6929.7	6928.6	6929.2	6928.4	6928.5	6928.8	6928.9	0.50
0.040	4591.1	4591.0	4591.2	4590.9	4591.0	4591.0	4591.0	0.10
0.036	2200.8	2200.1	2200.3	2200.1	2200.1	2199.9	2200.2	0.31
0.031	3211.9	3211.9	3212.0	3211.8	3211.9	3211.9	3211.9	0.07
0.031	9182.0	9181.5	9181.7	9181.6	9181.5	9181.5	9181.6	0.20

* Feature importance calculated from DT. # Standard deviation

**Table 2 pathogens-11-00586-t002:** The classification performance of four ML models using different feature selection strategies.

Only Use Top 15 Important Features to Classify		SVM	DT	RF	PR
Training	Accuracy	0.9135	0.9999	0.9999	0.7900
	Sensitivity	0.8687	1.0000	1.0000	0.6475
	Specificity	0.9432	0.9998	0.9998	0.8849
Testing	Accuracy	0.7261	0.6943	0.7509	0.7297
	Sensitivity	0.6337	0.6378	0.6453	0.5852
	Specificity	0.7906	0.7329	0.8250	0.8298
**Only Use All Important Features to Classify**		**SVM**	**DT**	**RF**	**PR**
Training	Accuracy	0.9485	1.0000	1.0000	0.8198
	Sensitivity	0.9184	1.0000	1.0000	0.7083
	Specificity	0.9686	1.0000	1.0000	0.8939
Testing	Accuracy	0.7377	0.7022	0.7666	0.7373
	Sensitivity	0.6530	0.6421	0.6551	0.6160
	Specificity	0.7968	0.7437	0.8455	0.8217
**Use All Features to Classify**		**SVM**	**DT**	**RF**	**PR**
Training	Accuracy	0.9939	1.0000	1.0000	1.0000
	Sensitivity	0.9897	1.0000	1.0000	1.0000
	Specificity	0.9967	1.0000	1.0000	1.0000
Testing	Accuracy	0.7522	0.6993	0.7241	0.6254
	Sensitivity	0.6686	0.6420	0.5688	0.5997
	Specificity	0.8121	0.7384	0.8326	0.6460

**Table 3 pathogens-11-00586-t003:** The classification performance of 4 ML methods with or without optimizing parameters.

Based on Optimal Parameter Settings		SVM	DT	RF	PR
Training (366 S/N samples)	Accuracy	0.8689	0.8470	0.9563	0.7732
	Sensitivity	0.8707	0.7959	0.9320	0.5850
	Specificity	0.8676	0.8813	0.9726	0.8995
Independent Testing (182 S/N samples)	Accuracy	0.8736	0.7692	0.7637	0.7582
	Sensitivity	0.7500	0.7222	0.6806	0.5139
	Specificity	0.9545	0.8000	0.8182	0.9182
	AUC	0.8664	0.7956	0.8244	0.8238
**Based on Default Parameter Settings**		**SVM**	**DT**	**RF**	**PR**
Independent Testing (182 S/N samples)	Accuracy	0.5879	0.7363	0.6538	0.5385
	Sensitivity	0.0972	0.6389	0.5972	0.6667
	Specificity	0.9091	0.8000	0.6909	0.4545
	AUC	0.5910	0.7170	0.7134	0.5427

**Table 4 pathogens-11-00586-t004:** The classification accuracy against MRSA with different types of SCC*mec*.

SCC*mec*	N	SVM	RF	DT	PR
Part 1: The best models optimized by clinical samples					
II	10	0 (0%)	0 (0%)	1 (10%)	0 (0%)
III	50	36 (72%)	43 (86%)	47 (94%)	37 (74%)
IV	23	9 (39%)	9 (39%)	7 (30%)	13 (56%)
V	27	11 (40%)	14 (51%)	13 (48%)	15 (55%)
Total	110	56 (51%)	66 (60%)	68 (62%)	65 (59%)
Part 2: Manually elevate the importance of feature 2410 to 2417					
II	10	10 (100%)	9 (90%)	10 (100%)	9 (90%)
III	50	45 (90%)	32 (64%)	50 (100%)	44 (88%)
IV	23	23 (100%)	16 (69%)	23 (100%)	22 (95%)
V	27	27 (100%)	17 (62%)	27 (100%)	26 (96%)
Total	110	105 (95%)	74 (67%)	110 (100%)	101 (92%)

## Data Availability

The summary data supporting the findings of this study are available within the article or its Appendix A. The data for individual patient subjects are not publicly available due to privacy/ethical restrictions.

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
