# Peer review of "Discrimination of Methicillin-resistant Staphylococcus aureus by MALDI-TOF Mass Spectrometry with Machine Learning Techniques in Patients with Staphylococcus aureus Bacteremia"

_pathogens, 2022, doi:10.3390/pathogens11050586_

Round 1

Reviewer 1 Report

The manuscript presented for review is well-written and structured report about discrimination of methicillin-resistant Staphylococcus aureus by MALDI-TOF mass spectrometry with machine learning techniques in patients with Staphylococcus aureus bacteremia.

Methicillin resistant Staphylococcus aureus strains pose a serious treatment problem. Therefore, the subject of presented manuscript is very important. The research was conducted in a very detailed way and the results are clearly presented.

The manuscript is interesting, complete and well structured, but it has some errors, and it is necessary to do a major revision to be accepted:

Line 24: no à not

Line 41 and 42: there are no relevant literature references to the cited works (Kumar et al., Seymour et al.)

I also did not find in the references: Huang et al., Wang et al, Tang et al. which are cited in the work. Every work cited in the text should be included in the references. Please correct this throughout the text

Figure 1: Data copied from Exel should be put into a separate table to improve the quality of this figure.

Line 143: set à set, and

Line 394: intensities, and à intensities and

Line 424: Secondly, removing a variable causes the Akaike Information Criterion (AIC) [52] to drop the most. à Secondly, removing a variable causes the Akaike Information Criterion (AIC) to drop the most [52].

Line 428: similar to à like

Line 430: larger and à larger, and

Line 449: were similar to à were like

Author Response

Responses for reviewer's comments are written in the attached file. Please see the attachment.

Reviewer 2 Report

The paper by Kong et al. describes the use of MALDI-TOF MS combined with machine learning to characterize bloodstream Staphylococcus aureus isolates as methicillin-resistant.  This paper discusses how the combination of MALDI-TOF MS combined with machine learning can predict whether a strain is a MRSA.  The work is certainly a step in the right direction.  Some points to consider.

  1. No banked MRSA strains were used that represent different SCCmec and spa types.  If developing a system, this would be a logical thing to do.
  2. How were the S. aureus isolates confirmed to be S. aureus?  Provide a methodology in the materials and methods.
  3. Provide a reference for the antibiotic susceptibility testing.
  4. How does your technology do against the different SCCmec and spa types?  Is there equal diagnostic sensitivity and specificity across these types?
  5. Line 2 should be methicillin.
  6. Line 21  "but routine clinical antimicrobial"

Author Response

(The authors gave the same response as above.)

Round 2

Reviewer 1 Report

I would like to thank the Authors for thoroughly addressing the review comments.

Reviewer 2 Report

The authors have addressed my concerns.